# Creating Successful Student Learning Outcomes: The Case of Trinity University's Quality Enhancement Plan Entitled "Starting Strong"

**John R. Hermann**

Department of Political Science, Trinity University, One Trinity Place, San Antonio, TX 78260, USA; jhermann@trinity.edu

**Abstract:** Using Starting Strong as a case study, this article examines how four successful Student Learning Outcomes (SLO's) emerged and one was eliminated during the Quality Enhancement Plan's (QEP's) development process. In comparison to the one that was purged, the four successful SLO's had five commonalities: 1. Virtually unanimous support from the administration; 2. Wide acceptance of the SLO from the faculty and staff members working on the QEP; 3. A shared conception between the administration and faculty/staff of what is an appropriate SLO; 4. The SLO's could be clearly conceptualized and measured; And, 5., the SLO's are financially feasible for the university to implement. The study hopes that this article may provide guidance for other universities undertaking and developing SLO's and QEP's.

**Keywords:** student learning outcomes; accreditation; quality enhancement plan; pedagogy; academic advising; academic support resources

---

## 1. Introduction

Trinity University is a small liberal arts college with a few select graduate programs located in the historic Monte Vista area in San Antonio, Texas. Trinity recently celebrated its 150th anniversary and has a total enrollment of approximately 2400 students. It advertises a nine-to-one student/faculty ratio. It has Presbyterian roots, but since 1969 it has been an independent secular university. For almost a quarter of a century, Trinity has been consistently ranked first in the western region among universities offering undergraduate and master's degrees.

After the financial crisis in 2008 in the United States, institutional data revealed that Trinity needed to strengthen its approach in helping first-year students succeed. Trinity decided that it would use its Quality Enhancement Plan (QEP) to make long-term sustainable changes to ensure that first-year students' transition from high school to college was more seamless. Since 2003, the Southern Association of Colleges and Schools: Commission on Colleges' (SACSCOC) reaffirmation of the accreditation process mandates that higher educational institutions within its region undertake and complete a QEP once every ten years. SACSCOC defines a QEP as a "topic that is creative and vital to the long-term improvement of student learning [that] . . . focuses on learning outcomes and/or the environment supporting student learning" [1] (p. 49). As the Chair of Topic Proposal Committee and Development Team at Trinity University, I worked on my university's QEP entitled *Starting Strong: Intentional Strategies for Improving First-Year Student Success* between 2015 and 2018. *Starting Strong* is designed to improve first-year teaching, advising, and academic support resources.

The QEP is predicated on student-learning outcomes (SLO's)—specific, "well-defined goals related to an issue of substance and depth, expected to lead to observable results" [1] (p. 49). While the success or failure of *Starting Strong* is dependent on many factors (e.g., faculty culture, ability to

fulfill the promises made, and how to work within the finite resources allocated to the QEP), SLO's may be the most challenging element in developing a successful QEP. This article examines *Starting Strong's* experience with SLO's during Trinity's QEP's Development Phase—the good and bad, the challenges, and the hope of providing improvement for Trinity's first-year teaching, advising, and academic support resources. It is important to note that Trinity will not know the influence of the SLO's until after QEP Impact Report in 2024.

This article provides a brief history of SLO's and discusses the benefits and criticisms of them. It also explores Trinity's experience with SLO's with the development of *Starting Strong*. In particular, it examines how four conceptually clear and measurable SLO's were created and why one was eliminated. The article provides a discussion of what was learned in Trinity's use of SLO's during the development of *Starting Strong*. It also discusses improvements that universities can make to help students succeed. The hope is that this article may provide guidance for other universities undertaking and developing SLO's and QEP's.

## 2. History, Benefits and Criticisms of SLOs

SLO's did not arrive as an orphan on the doorstep of higher educational institutions in the middle of the night. The concept was borrowed from a business practice in the 1980s called Total Quality Management (TQM)—"a management approach to long term success through customer satisfaction" [2]. TQM held businesses accountable on eight dimensions: Customer-focused, total employee involvement, process-centered, integrated system, strategic and systematic approach, continual improvement, fact-based decision making, and communication. All of the concepts in TQM are supposed to be clearly conceptualized and measured [2].

Borrowing from the TQM conceptual framework in the 1990s, Outcome-Based Education (OBE) was introduced in higher education to ensure that "college courses ... have expressed measurable results" [3] (p. 2). SLO's are the direct descendant of OBE's. Both SLO's and OBE's are designed to "apply business methods and values to higher education" [3] (p. 2). While many in liberal arts disciplines are skeptical of applying business concepts to the academy, there is no doubt that SLO's may have at least three benefits. First, a college education is exorbitantly expensive. Prospective students and their families have a right to know that they will receive a return on their investment. SLO's are a way of keeping higher education institutions accountable to their consumers, government regulators, and banks that provide loans. In the United States, student loans are not forgivable in bankruptcy court, because it is secured debt. Second, SLO's require that universities in the SACSCOC's region be involved in self-evaluation in the hope of continual improvement. As a form of assessment, SLOs aspire to develop "more sophisticated understandings of their students and their campuses but also [to create] a number of powerful interventions to improve learning" [4] (p. 117). Third, although frequently viewed as simply part of the compliance process, SLO's may improve the purpose of accreditation. Prior to the introduction of SLO's, the focus of accreditation was measuring resources (e.g., how many volumes a library has or what percentage of faculty hold a PhD or terminal degree in their field). These are indirect measures of student learning. By comparison, SLO's attempt to focus on teaching and learning effectiveness through observable and desirable results [5] (p. 15). Teaching and learning are supposed to be direct measures of universities acting in the students' best interest. If SLO's achieve their objectives, there can be sustainable change in learning.

While SLO's have benefits, they also have been criticized. Clemens [3] argues that SLO's are directly related to the decline of American education. He believes that SLO's are used by "accreditors to control institutions, and institutions control faculty and curriculum" [3] (p. 2). His argument strongly implies that accreditors and institutions are infringing on faculty autonomy through the use of SLO's. Clemens also points out that, "Learning is personal, internal, and the result of many complex interactions [3] (p. 4). Learning is not always measurable. It also takes time, not the result of one course or one exam. Learning is a more nuanced, life-long process. Immediate learning may not be the best way to measure the benefits of an education.

Equally important, SLO's are difficult to measure because the burden of the student learning is frequently placed on the faculty. The student is equally responsible for learning. A faculty member can be a brilliant teacher and use innovative pedagogical practices. Yet, it does not guarantee that the student will learn the material in class. If the SLO does not show measurable progress, does the blame lie with the faculty member, student, or university? While the use of SLO's has the intention of keeping universities accountable for providing a first-rate education, it is not always the case. Other factors may be at play.

### 3. Starting Strong and SLOs

Benevolent or nefarious, SLO's are here to stay. Universities operate in an assessment culture. As Felten and colleagues point out: "Indeed, at some institutions the assessment director is like a modern-day Paul Revere, riding through campus to raise the alarm, "The accreditors are coming! The accreditors are coming!" [4] (p. 117). My experience in leading *Starting Strong* was no exception. The goal of the QEP was approval by SACSCOC's without any conditions.

*In Starting Strong's Case, We Developed Five SLO's:*

1. First-year students will demonstrate an understanding of Pathways, registration procedures, and requirements for graduation.
2. First-year students will explore their academic, career, and life goals.
3. First-year students will be able to assess their academic performance by the fifth week of class to identify areas that need improvement.
4. First-year students will identify institutional resources that help them overcome academic challenges.
5. First-year students will demonstrate help-seeking behaviors to reach their academic goals.

With the exception of the second one, the Development Team approved all of the SLOs after being proposed by the Teaching, Advising, and Academic Support Resources subcommittees. We also had a Steering Committee, which included the leaders of the subcommittees. They communicated with one another about the progress and challenges of each subcommittee's work, including their work on the SLO's. In consultation with the Board of Trustees, the President of Trinity and his Executive Council also approved all of the SLO's. In the end, SACSCOC's approved the four SLO's and QEP without any conditions.

The approved SLO's had five commonalities. First, they had virtually unanimous support from the administration. Second, they were widely accepted by faculty and staff members working on *Starting Strong*. Third, there was a shared conception of an appropriate SLO between the administration and faculty/staff, that is, they were consistent with Trinity's culture. Fourth, the approved SLO's could be clearly conceptualized and measured. Fifth, they were financially feasible for Trinity to implement.

Some of the SLO's were easier to develop than others, because they were directly related to student learning. This was the case with SLO's 1 and 3. For example, SLO 1 seeks for students to understand our curriculum (i.e., Pathways), registration procedures, and the basic requirements for graduation. SLO 1 involves transactional advising—an actor at Trinity transfers key information to the first-year student. The central strategy to accomplish SLO 1 was to hire an Advising Coordinator. In consultation with the Advising and Registration Committee, Academic Affairs, and Chairs of Departments, the Advising Coordinator would standardize the information students receive about first-year advising. It would ensure that there is a certain quality standard, minimizing the possibility of variation in the first-year students' knowledge of advising and registration information. Two major documents are distributed to all first-year students: The Pathways (the name of Trinity's curriculum) evaluation form and the First-Year Course Guide. The Pathways curriculum evaluation is a one-page summary of Pathways' requirements. The First-Year Course Guide provides suggested courses in each department for first-year students and which classes fulfill certain Pathway requirements.

Similarly, SLO 3 was relatively easy to create and develop, because it could show a clear benefit to student learning. Through small syllabi revision grants, faculty are encouraged to use low stakes assignments early in the semester and use "early alerts" (after the fifth week of class) to notify students any areas where they need improvement. There is scholarly consensus (e.g., [6] (pp. 55, 57), [7] (p. 103)) that notifying first-year students of deficiencies after midterms may be too late. Members of the Teaching Subcommittee and the Development Team believe that first-year students would benefit from more explicit feedback from their professors earlier in the semester. First-year students may need help interpreting or reacting to instructor feedback with behaviors that support student learning. SLO's 1 and 3 provide clear guidelines on how to help students change their behavior in advising and teaching. The student-learning outcomes can also be measured.

While more challenging than creating SLO's 1 and 3, SLO's 4 and 5 were approved by the Development Team and the President and his Executive Council in consultation with the Board of Trustees. SLO's 4 and 5 dealt with knowledge and the use of academic support resources by first-year students. Through various strategies, Trinity could show a connection between the knowledge of and the use of academic support resources to improve students learning. The connection between the strategy and the improvement of student learning is vital in the development of the SLO.

SLO 4's central aim is to provide first-year students with knowledge of the institutional resources available to them when they face academic challenges. This is accomplished in two ways. First, Starting Strong uses strategic marketing by the Student Success Center to faculty members and first-year students about the academic support services available (e.g., the dissemination of information to first-year students and presentations in Trinity University's Teaching and Learning Collaborative for faculty). Second, *Starting Strong* decided to strengthen its academic support resources. Recognizing that many of Trinity's gateway classes were in STEM, we added more tutors, introduced Supplemental Instruction (SI), adopted a software package called ALEKS for our Calculus class, and hired a director for the Quantitative Reasoning and Skills Center. At the heart of the Center will be mathematics placement and tutorial software using artificial intelligence to help students bridge the gap between what they have mastered in high school and what is needed for success in a quantitative college course. The Quantitative and Reasoning Skills Director works with academic departments, Academic Support staff, and Information Technology Services staff to ensure that these needs are met. The Director will also hire, support, and train existing and new tutors in the STEM field.

SLO 5 is the most ambitious in terms of what we expect students to do differently. Our goal is for students to demonstrate help-seeking behaviors in order to meet their academic goals. It is one thing for first-year students to forge strong relationships with first-year advisers, receive earlier and clearer feedback on graded work in first-year classes, and have access to exemplary academic resources. It is another for first-year students to be proactive in their early academic careers. First-year students need to learn when to visit their advisers and faculty and, more importantly, what questions to ask. First-year students need to know when and where to seek academic support resources. A one-size-fits-all approach will not work. SLO 5 may be the most essential to long-term academic and career success. Yet, it asks the student to transition from being semi-dependent to independent in resolving issues.

SLO 5 requires that Trinity have the academic support resources to resolve issues. After all, students cannot engage in help-seeking behavior if there is not a resolution for the students. It also requires the Director of Academic Support and the Associate Vice President for Academic Affairs: Student Academic Issues and Retention will work with Strategic Communications and Marketing staff to develop and implement a marketing plan with the goal of publicizing academic support resources and designating their use. The Director of Assessment plans to send surveys to first-year students asking a series of questions to gauge whether students were engaged in help-seeking behavior.

## 4. The Purged SLO

SLO 2 was the only one eliminated after being approved by a Development Team subcommittee. SLO 2 was created to build a relational dimension on the transactional element of SLO 1 (i.e.,

understanding Pathways, registration procedures, and graduation requirements). In its original form, the Advising subcommittee proposed SLO 2 as: First-year students will have higher quality interactions with their advisers. It was designed to strengthen advising by making it relational – the adviser needs to demonstrate good listening skills, convey an attitude of warmth and welcome, and ask questions that invite the student's involvement in the discussion. O'Bannion [8] goes one step further by creating a hierarchy of advising (from relational to transactional):

1. Exploration of life goals, values, interest, aptitudes and limitations
2. Exploration of career goals consistent with the student's goals and values
3. Choice of program, major, and minor
4. Course selection
5. Scheduling classes

SLO 2 was predicated on Richard Light's observation that, "good advising may be the single most underestimated characteristic of a successful college experience" [9] (p. 81).

SLO 2 ran into conceptual and measurement issues when the Development Team and those involved in assessing the QEP reviewed it. What does it mean by having "higher quality interactions with advisees?" Does it mean first-year advisers will subscribe to a particular advising philosophy (e.g., appreciative advising, Socratic advising, developmental advising, or intrusive advising)? How will we train the faculty in these advising techniques? Are faculty members willing to make the needed changes without extrinsic incentives?

Some members of the Development Team revisited the wording of SLO 2. The amended change was: First-year students will explore their academic, career, and life goals. Even with the change, SLO 2 met resistance. Many faculty members believed it was beyond their job description to engage in relational advising. Others thought it was a good option to allow faculty to make a choice as to whether they wanted to engage in discussing academic, career, and life goals. The administration was willing to pay for a small grant for first-year advisers to attend a workshop where they were trained in elements of relational advising in addition to transactional advising, but SLO 2 should not be mandated. Those assessing *Starting* Strong also found it challenging to create a reliable and valid measure for SLO 2. The result was that SLO 2 was dropped from the QEP. Even though student engagement may be the most important determinant of student success in college [10], Trinity was not ready to take the step of mandating faculty to engage in relational advising.

## 5. Discussion

Support from the administration, faculty, and staff are vital for creating and developing successful SLO's and a QEP. It requires those working on QEP's to understand the nuances of the university culture—much easier said than done. It is prudent to stray from lightning rod issues. SLO's need to be clearly conceptualized and measured. Finally, for the successful implementation of the SLO, financial and personnel resources must be available. While there is only a short list of requirements to create a successful SLO, they can be surprisingly difficult to meet with so many different constituencies and conflicting needs on a college campus.

There are certain strategies that all universities can take to improve student learning and success. Additionally, not all of them involve an exorbitant amount of resources. Faculty members are seen as the front line in educating students. While much learning transpires outside of the classroom, the way faculty approach student learning may be most important. In the course of obtaining an advanced or terminal degree, most faculty members are not trained in pedagogy. Faculty members are certainly experts in their fields and can convey that information to students. Yet, less time is devoted to teaching students how to learn. Understanding material and student learning are frequently different. Yet, if the focus is on student learning, it can bring the greatest dividends to students and the university. SLO's can help achieve greater focus on pedagogy.

Virtually all syllabi clearly explain the course material, the graded work, and the sources consulted in class. A few pedagogical additions can be the difference between the success and failure of the student in class. Learning objectives for the class can offer a student a roadmap of the expectations for the student. Specific skills learned and why they are valuable will help improve how the student learns and professor teaches. For example, if there is a paper in the class, the faculty member can explain what a student will learn in the paper. A learning objective can be as simple as creating a clear and succinct central claim. The faculty member can also show what a good example of a central claim is and what is not. Learning objectives can also be used as a measurement tool for assessing the student's growth in the class.

Universities also frequently suffer from "siloization". Departments, offices, and administrators all have defined responsibilities. The departments, offices, and administrators work effectively in their delineated duties. It has clear benefits, but it often precludes clear communication among university actors in other areas. Student Success and Financial Aid offices, Student Life offices, faculty members, advisers, and administrators need to communicate to ensure the student is doing well in all areas of her life. A student facing challenges in one area of her life is more inclined to suffer in others. Student success takes a village of university actors to act in the student's best interest.

"Early alerts" (usually after the fourth or fifth week of class) with low stakes assignments are helpful in combatting any issues the students are having in class, especially compared to waiting until midterm grades are released. Still, the information conveyed in early alerts is essential to student success. In one of the sections of our First-Year Experience, two faculty members who co-taught together met with each student in the class. The faculty members discussed what the students were doing well, how they could improve, and what help-seeking strategies they could engage in to remedy any issue. After the meeting, the student had one week to respond by email to reflect on the discussion. According to the faculty and students in the class, it was the meaningful dialogue that led to greater student success in that particular class as well as other classes the students were taking that semester.

Finally, with the creation of SLOs, it is vital to consider who is going to do what. Additionally, how that new task will influence the employee's current workload in her current position. If the employee works full time, it is assumed that the workload is already taken up with essential tasks. Thought and preparation must go into creating new responsibilities. If it is a new position, new office space needs to be created to ensure student success. This also includes professional development to ensure existing and new employees are kept current on best practices.

SLO's are an essential part of university culture. We need to embrace and leverage them to benefit the students' best interest. SLO's are more than a compliance issue. Instead, SLO's should enhance the learning culture in a sustainable way. This will not only help in the accreditation process. It will also benefit the student in the learning process.

**Author Contributions:** J.R.H. contributed to the research and writing of the article. The author have read and agreed to the published version of the manuscript.

**Funding:** This research received no external funding.

**Conflicts of Interest:** The author declares no conflict of interest.

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
