# Peer review of "Creating Successful Student Learning Outcomes: The Case of Trinity University’s Quality Enhancement Plan Entitled “Starting Strong”"

_sustainability, doi:10.3390/su12198197_

Round 1
Reviewer 1 Report
The paper present very interest perspective on student learning outcomes. It is written in a popular way of writing. It is more suitable for blog.
The referencing is poor and not adequate.
Author Response
Peer Reviewer 1:
Of all the reviewers, Peer Reviewer 1 asked me to make the most substantial changes to my article-length manuscript. In particular, it seems that Peer Reviewer 1 perused my article as a research-based article with new data. Instead, the central focus of my article is a discussion. Thus, the comments about the research design, methods, and findings are not applicable. My article focuses on the creation and development of SLO’s in an applied environment, so the comments seem incongruent with the purpose of my article.
Peer Reviewer 1 also states that my article is “written in a popular way of writing. It is more suitable for a blog.” Since I do not read many blogs, I am not sure what Peer Reviewer 1 means. The other peer reviewers, moreover, complimented my writing for its clarity. It has been my objective to improve my writing by making it as accessible as possible to a wide audience, not to a narrow scholarly community. Since the other three reviewers expressly mention that my discussion is well written, clear, and appropriate.
Finally, Peer Reviewer 1 states that my “referencing is poor and not adequate.” I am not sure where in the article the referencing is poor and not adequate and where it is adequate. I can assure the editors that I did a fairly exhaustive search of the appropriate literature on SLO’s, QEP’s, and first-year students for the article. Once again, this article is a case study based on an applied setting. The literature review was a large part of the QEP, which I wrote.
Reviewer 2 Report
Creating Successful Student Learning Outcomes: The 3 Case of Trinity University’s Quality Enhancement 4 Plan entitled “Starting Strong”
Concise and straightforward, but would help to set the geography: Where is Trinity University? How large is it? What kind of an institution is it: undergraduate, grad, public, private?
Very strong summary of pros and cons of SLOs.
Line 184. Shouldn’t it read, “First-year students will have higher quality interactions with their advisors (not advisees.)”
Overall, the paper is extremely well written and interesting. On the other hand, why is it being submitted to this journal? Do these SLOs have anything at all to do with sustainability? If so, the author needs to make that clear. Moreover, the manuscript feels like an opinion piece as no real data has been collected to suggest whether the adoption of these SLOs has any bearing on the outcome of students.
Author Response
Peer Reviewer 2:
Unlike Peer Reviewer 1, all of Peer Reviewer 2’s comments and suggestions were adopted and implemented in the manuscript.
Peer Reviewer 2 asked for more information on Trinity University, which helps understand the context of the article. I have incorporated Peer Reviewer 2’s suggestions (location, size, type of college, and on and on) in lines 22-27 of the manuscript. Here is what I wrote:
“Trinity University is a small liberal arts college with a few select graduate programs located in the historic Monte Vista area in San Antonio, Texas. Trinity recently celebrated its 150th anniversary and has a total enrollment of approximately 2400 students. It advertises a 9-to-1 student/faculty ratio. It has Presbyterian roots, but since 1969 it has been an independent secular university. For almost a quarter of a century, Trinity has been consistently ranked first in the western region among universities offering undergraduate and master’s degrees.”
Peer Reviewer 2 also makes a great point that I do not create a linkage between my article and the issue of sustainability. To remedy this issue, I have included a couple of sentences in lines 28-31 on the nexus between my article and sustainability. I also did the same on lines 91 and 291 of the article. Based on this suggestion, my article is a better fit for Sustainability.
Peer Reviewer 2 also caught a mistake on line 184 where it should read advisees (not advisers). I have made the correction. And, thank you!
Finally, Peer Reviewer 2 makes another excellent point that we will not know the influence the SLO’s actually have on the students. I could not agree more. It is definitely a limitation of the study. I did add a cautionary sentence that Trinity will not know the influence of the SLO’s until the QEP Impact Report in 2024 (lines 55-56). On the other hand, QEP’s must be well-researched, meaning that it needs to have an extensive literature review, incorporation of best practices, and a comparative analysis to peer and aspirant institutions. The established measures provide guidance in creating and developing the SLO’s, although the past is not always the best predictor of the future.
Reviewer 3 Report
This was the discussion paper on programes promoting students learning outcomes. Overall I liked the paper, it was clearly and interestingly written.
Author Response
Peer Reviewer 3
Peer Reviewer 3 is the most complimentary of my article and does not make suggested changes. But, I want to thank Peer Reviewer 3 for stating that she “liked the paper, it was clearly and interestingly written."
Reviewer 4 Report
Overall, this is a well written manuscript with a cogent and compelling argument established and maintained throughout. There are a few areas that require attention:
Abstract
Consider rephrasing, “the article examines why four…” The article doesn’t answer the question “why?”, it answers the question, “how”.
Section 2
line 63 – the statement “Student loans are not forgivable in bankruptcy court, because it is secured debt” is not relevant in countries outside the author’s country. It would be good to add a qualifier.
line 64 – statement, “Second, SLO’s require that universities be involved in self-evaluation in the hope of 64 continual improvement.” While this might be the case, there are many universities across the globe, in Australia and various provinces in Canada, that have required self-evaluation long before SLO’s came about. I recommend the author add some qualifiers, rather than making universal generalized statements.
The revisions I have recommended are minor, as overall, this is a strong submission.
Author Response
Peer Reviewer 4:
Like Peer Reviewer 2, I adopted all of the suggested revisions by Peer Reviewer 4.
Peer Reviewer 4 asked for me to revise line 8 by changing the wording from “why” to “how.” It never occurred to me that I was answering the “how” question. But, after seeing the suggestion, Peer Reviewer 4 is correct.
Additionally, Peer Reviewer 4 points out in two places where I need qualifiers in my article. I made the suggested corrections by:
- On line 81, I added that student loans are secured debt in the United States, implying that it may not be the case in other countries, which I did not know prior to Peer Reviewer’s 4 comment on the issue.
- According to Peer Reviewer 4, universities in other countries outside of the United States have been required to engage in self-evaluation before SLO’s burst on the scene. My correction narrowed the self-evaluation process of SLO’s to the SACSCOC’s region of the United States (See Lines 81-82).
Finally, consistent with Peer Reviewer 2, I discussed more appropriate background information in two ways.
- I provided a richer background of Trinity University in lines 22-27.
- I created a linkage between sustainability and my article in lines 28-31, 91, and 291.
Round 2
Reviewer 1 Report
There is still for improvement but it may be published as it is.
Reviewer 2 Report
The paper is a strong paper for what it is, but the author's perspective on Sustainability does not seem to match the understanding of the term used by members of the environmental community. A reader would expect some relevance to the UN SDGs, or at the very least reference to older notions of environment, equity, and economy, none of which seem to have much to do with the article.
The article deserves publication, but probably not in this journal.